# A redesign of OGC Symbology Encoding standard for sharing cartography

Erwan Bocher[1],* and Olivier Ertz[2],*

[1] CNRS, Lab-STICC Laboratory, UMR 6285, Vannes, France
[2] Media Engineering Institute, HEIG-VD, University of Applied Sciences and Arts Western Switzerland, Yverdon-les-Bains, Vaud, Switzerland
* These authors contributed equally to this work.



## ABSTRACT

Despite most Spatial Data Infrastructures offering service-based visualization of geospatial data, requirements are often at a very basic level leading to poor quality of maps. This is a general observation for any geospatial architecture as soon as open standards as those of the Open Geospatial Consortium (OGC) are applied. To improve the situation, this paper does focus on improvements at the portrayal interoperability side by considering standardization aspects. We propose two major redesign recommendations. First to consolidate the cartographic theory at the core of the OGC Symbology Encoding standard. Secondly to build the standard in a modular way so as to be ready to be extended with upcoming future cartographic requirements. Thus, we start by defining portrayal interoperability by means of typical-use cases that frame the concept of sharing cartography. Then we bring to light the strengths and limits of the relevant open standards to consider in this context. Finally we propose a set of recommendations to overcome the limits so as to make these use cases a true reality. Even if the definition of a cartographic-oriented standard is not able to act as a complete cartographic design framework by itself, we argue that pushing forward the standardization work dedicated to cartography is a way to share and disseminate good practices and finally to improve the quality of the visualizations.

## INTRODUCTION

Given how good geospatial technologies take advantage of the constant evolution of information and communication technologies, Spatial Data Infrastructure (SDI) appeared as a new paradigm in geospatial data handling. It extends desktop GIS (*Craglia, 2010*) where data collected by other organizations can be searched, retrieved and manipulated for several usages (*Tóth et al., 2012*). Many regional, national and international initiatives have setup well-defined access policies to promote the arrangement of SDI because location information is important in managing everything that a governance has to organize.

Currently, several SDI initiatives are particularly well implemented to encourage data discovery and sharing across different communities with various applications. Also service-based visualization of geospatial data is part of the SDI components. In the case of

Corresponding author
Olivier Ertz, olivier.ertz@heig-vd.ch

INSPIRE, the infrastructure for spatial information in Europe, requirements are defined at a basic level according to *INSPIRE Drafting Team (2014)*, in section 16, and *INSPIRE Drafting Team (2008)* in section A.11, which defines only general portrayal rules as recommendations. As an example, we may notice the technical guidelines on geology (*INSPIRE Thematic Working Group Geology, 2013*) which does not specify styles required to be supported by INSPIRE view services (section 11.2) but only recommended styles, often simple to excess, just defining some color tables, stroke width and color, spacing for dashed lines and graphic patterns to repeat in a surface or over a line. These are relatively simple to render with current implementation standards in use. Extreme simplicity may be intentional for some cases, but it may also reveal limitations from these implementation standards as soon as styles resulting from a cartographic design are more complex (*Ertz, 2013*). As a consequence, according to *Hopfstock & Grünreich (2009)*, with cartographic rules defined at such a basic level, portrayal seems to be considered as a concern of second zone, almost ignoring "the importance of visualization for transforming spatial data into useful GI." Even worse, some contemporary maps coming from SDI exhibit a serious lack of knowledge in cartography with many map-makers repeating some basic mistakes. Such as maps from *Eurostat/Regional Statistics (2017)* where population is represented as a choropleth map (e.g., population on 1st of January in NUTS 2 regions). *Field (2014)* points out that the current demand is for quantity, not for quality, and it is the Internet (not the discipline of cartography) which is reacting to this demand.

*Hopfstock & Grünreich (2009)* underline that poor map design results are the consequence of a "too technology- and/or data-driven approach" and propose improvements by making the cartographic design knowledge explicit and operational. Beside such a relevant proposition at the design level, this paper has a focus on the implementation level by making portrayal interoperability operational through the improvement of the open standards dedicated to cartography. Indeed, interoperability is key for SDI as interconnected computing systems that can work together to accomplish a common task. And the presence of open standards is required to allow these different systems to communicate with each other without depending on a particular actor (*Sykora et al., 2007*). The common task presently in question is about the ability for a user community interconnected by interoperable systems to share a cartography used for the authoring of a map. That is, not only the result of a cartographic rendering built of a set of pixels, but also the underlying cartographic instructions which describe how the map is authored. We can figure out how such an ability would participate to empower all types of users, from the cartographic professionals to data artists, journalists and coders (*Field, 2014*) to gain useful geographical information by means of cartographic visualizations. An ability that contributes to the power of maps, from tools which enable the sharing of spatial information and knowledge, to collaboration through shared creativity and skills transfer between "produsers" for better decision making (*Bruns, 2013*).

For cartographic portrayal interoperability, many SDI policies, like *INSPIRE Drafting Team (2014)*, advise the use of standards from Open Geospatial Consortium (OGC) like the Styled Layer Descriptor (SLD) (*Lupp, 2007*) and Symbology Encoding (SE)

specifications (*Müller, 2006*), but it seems these standards were not able to bring to reality the above vision that goes as far as considering SDI as open participation platforms. We might blame the fact that moving from closed monolithic applications to open distributed systems is still under way (*Sykora et al., 2007*) and that cartography must take effect providing a methodology with a user-oriented approach (*Hopfstock & Grünreich, 2009*). But this paper wants to show how it is also important to have syntactic portrayal interoperability operational with a mature open specification able to standardize the cartographic instructions. We show that the current OGC SE standard does offer limited capabilities for describing cartographic symbolizations. Then, while we develop some recommendations to improve the situation through more capabilities to customize the map symbology, we also propose some good practices to favor the adoption of the standard by implementors so as to make it really operational for the long term. We believe that these propositions should lead to rich cartographic portrayal interoperability, going further than basic styles. There is no reason SDI users have to be satisfied with often unsuitable maps.

## FROM MAP DESIGN TO PORTRAYAL INTEROPERABILITY

Clearly, many definitions and types of map exist. As *Tyner (2010)* writes "We all know what a map is, but that definition can vary from person to person and culture to culture." However, many of them do share the idea of a map as an intellectual construction that is based on the experience and knowledge of the cartographer to manipulate data input according initial hypotheses and its capacity to play with graphic signs (*Slocum et al., 2009*; *Tyner, 2010*). Furthermore, even if the definition is hard to settle, cartographers have also worked to formalize map syntactics by developing symbol categories and rules to combine them. Visual variables are symbols that can be applied to data in order to reveal information. Largely based on the *Bertin & Berg (2010)* classification, several cartographic authors agree with a set of commons visual variables (*Carpendale, 2003*; *MacEachren, 2004*; *Tyner, 2010*): shape, size, hue (color), value, texture, orientation (Fig. 1).

To create a map, they are individually manipulated or piled up by the cartographer in the process to visually map information about point, line and area features to visual variables (*MacEachren, 2004*; *Slocum et al., 2009*). This visual mapping is an embellishment design to improve the aesthetic quality and express efficiently a message (*Wood & Fels, 1992*). Even if creating map is an aesthetical exercise it is also a science that must respect some rules to make sure that the representation is accurate. A de facto set of best practices based on visual variables has been accepted by the academy of cartographers (*Montello, 2002*; *McMaster & McMaster, 2002*). As *Bertin & Berg (2010)* explains, the choice of the "right" visual variable, which would be most appropriate to represent each aspect of information, depends on the type of geographical object but also its characteristics (*MacEachren, 2004*; *Nicolas & Christine, 2013*). For example, like the statistical nature of the data (qualitative, quantitative), raw data must be represented with proportional symbols and a density of values by an areal classification (i.e., a choropleth map).

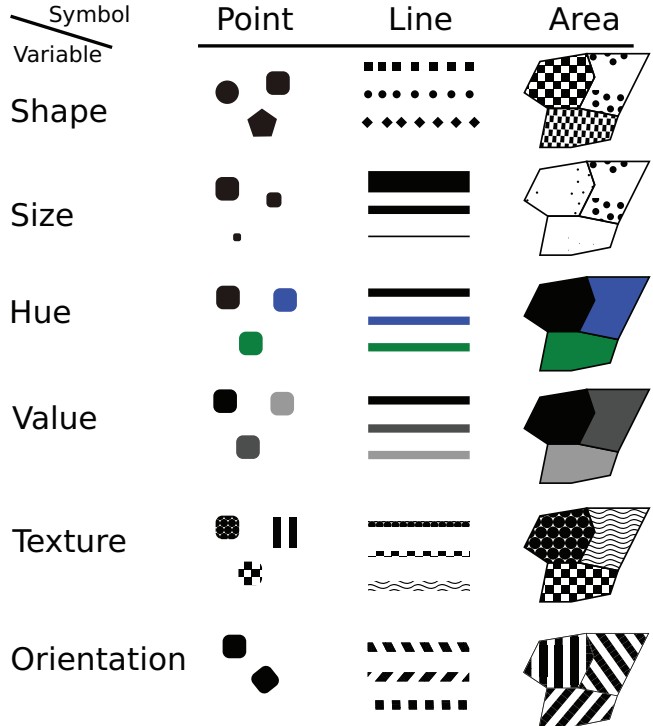

**Figure 1  The visual variables of symbols.**

These map syntactics are the results of the mainstream cartographic theory and the related design knowledge that help to understand how and why certain displays are more successful for spatial inference and decision making than others. This subject is an important issue to improve map quality at the design phase (*Hopfstock & Grünreich, 2009*). But also at the implementation phase, the theory related to these visual variables to compose map symbols is suitable to drive the definition of a standardized styling language that must be functionally designed and implemented into the geospatial tools making up SDI.

In order to explain how such a standardized styling language is an essential piece to enable cartographic portrayal interoperability, let us clarify the related concept of sharing cartography. We consider four use cases typical of sharing levels:

- **Level 1: discover**
  At this level, SDI users discover pre-styled and ready to be visualized map layers, eventually coming from different systems, they can combine to build a map. For example, it corresponds to the classical geoportal applications offering the user to discover and explore prepared maps and combine prepared layers from various thematics (e.g., map.geo.admin.ch). Typically, it does also match with the story of the fictive SDI user Mr Tüftel in the Web Portrayal Services book (*Andrae et al., 2011*). Mr Tüftel wants to unify on the same map the water pipes from his municipality but also the pipes from the municipalities in the neighborhood. These are different data sources he wants to combine in his everyday GIS tool. Finally, during the discovery of

some cartographic facets, the user gains knowledge of the potential of the underlying data sources hosted by the different systems.

- **Level 2: author**
  Starting from level 1, the potential of the underlying data sources may give to the SDI user some ideas of analytical process which requires to create a new style different from the default. For example, this is useful for Mr Tüftel in the case he would like to create an unified map of water pipes, but with the problem of getting different visualizations of the pipes (e.g., different colors) from the different municipalities. He would then author a common style (e.g., same color) so as to take the control of the whole rendering process. Even further, Mr Tüftel may enrich the analytical process and take benefit of an extra underlying data that classifies each pipe according to its function (either wastewater or rainwater). He would then author a new style (e.g., orange color for wastewater pipes, blue color for rainwater pipes) so as to produce a suitable map to decide where to build the intercommunal water treatment plant.

Starting from level 2 some specific use cases become relevant:

- **Level 3: catalog**
  It is about having at disposal style catalogs offering ready-to-use styles, often tailored for specific thematics, e.g., noise mapping color palettes (*EPA, 2011*). The ability to import such a specialized symbology into users' tool just avoid to reinvent the wheel in the sense of re-creating the style from scratch. By analogy, the catalog style use case is similar to how the OGC Catalog Service for metadata works.

- **Level 4: collaborate**
  The context of this use case is wider and involves several SDI users into a collaborative authoring process. Several users contribute to the creation of a common map, each user having specialized skills to complement one another so as to tell stories as maps, each using her(his) own software (*Ertz, Julien & Bocher, 2012*). In other words, cartographic portrayal interoperability enable the freedom to the users to work with the tools they are most comfortable and productive with. Also, we may notice the educational capacity of this use case. Considering a team of people with different levels of skills in cartography, there are offered the chance to share them.

As pointed out by *Iosifescu-Enescu, Hugentobler & Hurni (2010)*, "the use of standardized exchange languages is commonly considered as the most practical solution for interoperability especially when it is required to collate resources, like data, from various systems," but also when it is to take the control of a distributed cartographic rendering process. Definitely, starting from level 2, the definition of a standardized styling language is essential to share cartography: that is the underlying cartographic instruction, what we call the symbology code which constitutes a style that describes how a map is authored. Such a definition can be achieved in the same way *Iosifescu-Enescu & Hurni (2007)* try to define a cartographic ontology by considering that "the building blocks for digital map-making are the primary visual variables (color, opacity, texture, orientation, arrangement, shape, size, focus) and the patterns (arrangement, texture, and

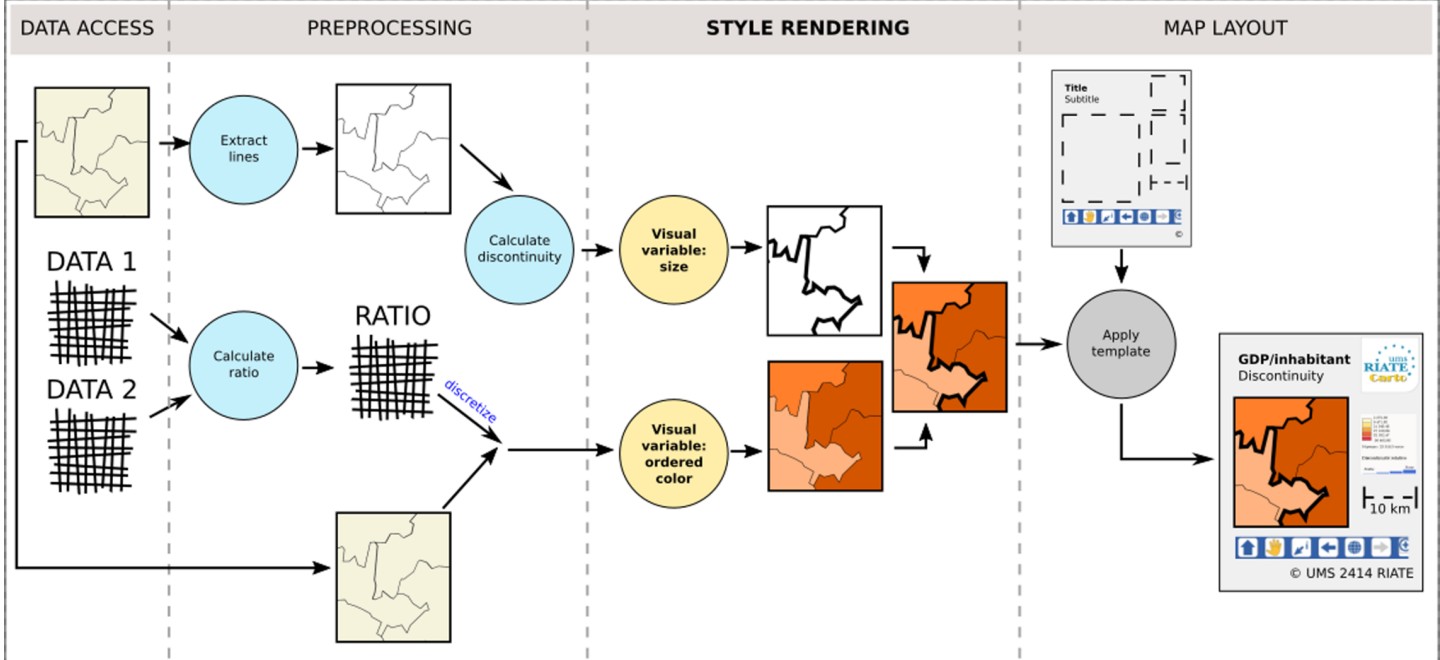

**Figure 2** **The four stages of the cartographic map design, inspired from** *Nicolas & Christine (2013)*.

orientation)." Also, another starting point is to consider a map (either general-purpose maps, special-purpose maps or thematic maps) as being composed of some graphic elements (either geometric primitives or pictorial elements). This approach matches the OGC SE standard which is the standardized styling language (*Lupp, 2007*) in question here: a style is applied on a dataset to render a map considering a composition of possible symbol elements (called Symbolizer) that carry graphical properties (equivalent to visual variables).

So as to complete the definition of cartographic portrayal interoperability, Fig. 2 shows that such a styling language is at the core of the third stage of the cartographic pipeline, the one dedicated to the style rendering. Thus it is to notice that the map layout design which configures a title, a legend, a north arrow, a scale bar, etc. (*Peterson, 2009*), is out of our scope, as well as the preprocessing stage which is dedicated to the preparation of the dataset to visualize. As an example, building an anamorphic map requires a preliminary processing to generate consistent geometries with preserved shape and topology before styling them.

The next part does focus on the technical aspects about how current open standards are able or not to fully meet the conditions of such a cartographic portrayal interoperability.

## OPEN STANDARDS FOR SHARING CARTOGRAPHY

Given the concept of sharing cartography defined by the above four use cases, let us see what are the possibilities and limits to implement them using OGC standards.

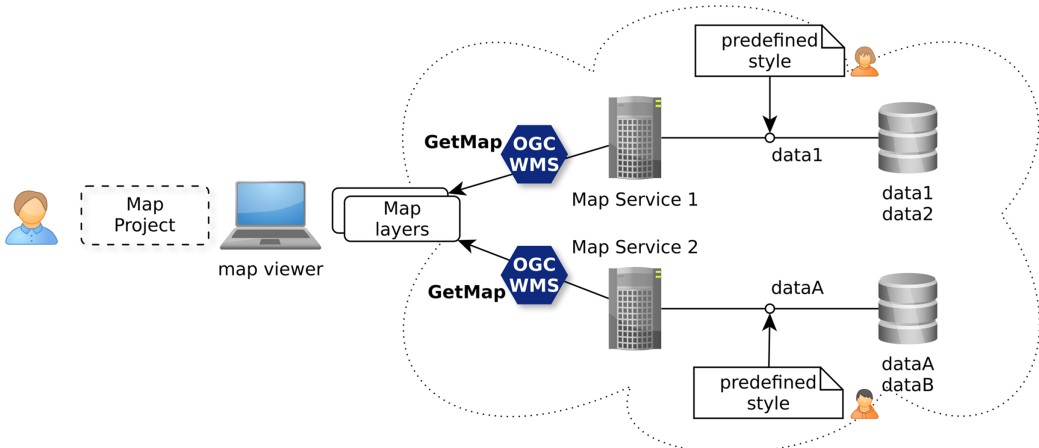

Figure 3 Discovery of ready to be visualized map layers with OGC WMS standard.

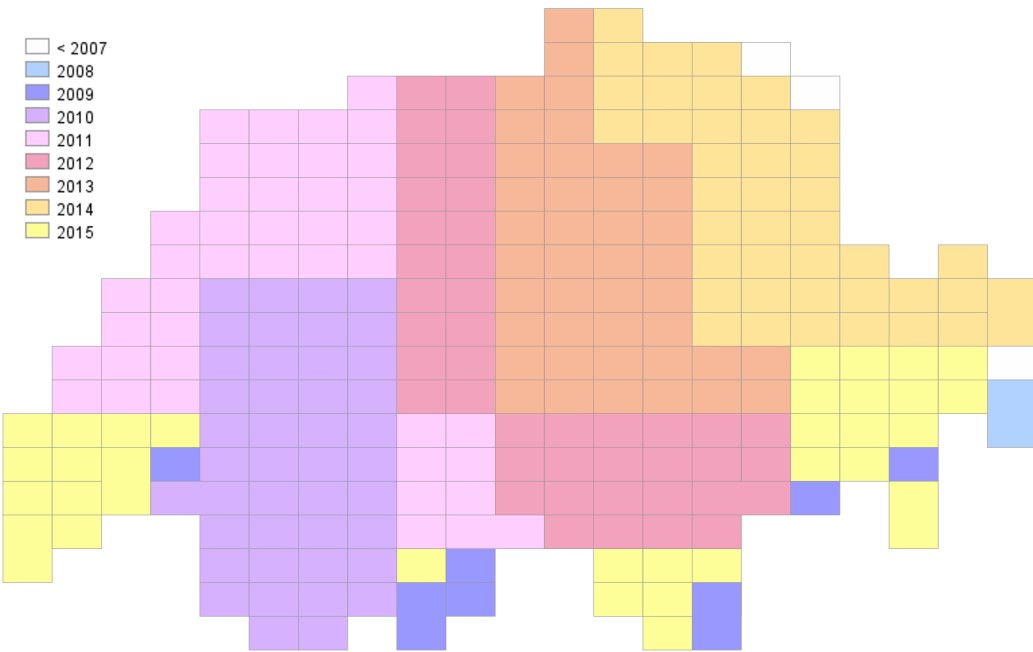

Figure 4 Visualization of the grid of map sheets of Switzerland (1:25,000) through a default cartographic style showing a choropleth symbology based on the year of edition of the sheet.

## Use case "discover"

The OGC Web Map Service (WMS) standard (*De la Beaujardiere, 2006*) is currently the only widely accepted open standard for map visualization which standardizes the way for Web clients to request maps with predefined symbolization (*Iosifescu-Enescu, Hugentobler & Hurni, 2010*). This ability, as illustrated with Fig. 3, does match the use case level 1 allowing to discover ready-to-visualize map layers and to combine them to build maps.

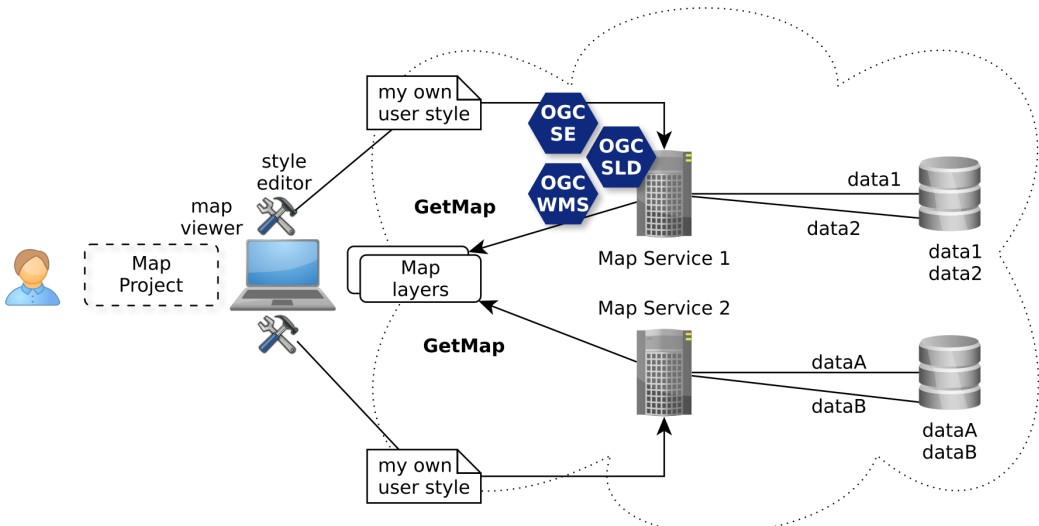

**Figure 5 Authoring of user style to visualize map layers with OGC WMS/SLD and SE standards.**

Just send a simple GetMap request to the Swisstopo WMS server to get a predefined colored map layer to overlay in your web mapping application (Fig. 4):

https://wms.geo.admin.ch/?SERVICE=WMS&VERSION=1.0.0&REQUEST= GetMap&FORMAT=image/png&LAYERS=ch.swisstopo.pixelkarte-pk25.metadata- kartenblatt&SRS=EPSG:21781&STYLES=&WIDTH=1895&HEIGHT= 1185&BBOX=475000,68000,854000,305000

The WMS GetMap operation allows to choose one of the internal styles prepared for a layer by a map-maker (parameter STYLES). Each style is related to one or more datasets attached to the WMS server and ready to be used by an end-user.

## Use case "author"

The analysis of the use case level 2 described in chapter 2 shows that it is required to establish an open framework able to facilitate decision making through customized maps. *Iosifescu-Enescu (2007)* does underline that the WMS standard combined with the SLD profile and the SE is able to fulfill such a requirement. The ability to drive remotely the authoring of visualizations is fundamental for this use case, for example to fulfill the cartographic requirements of Mr Tüftel. He does not want to download the spatial data, he just wants to adjust the visualization according to his specific needs (Fig. 5).

Just send the below WMS/SLD request which has a reference to a style file. This latter includes some SE instructions which allow to get a customized visualization (Fig. 6):

https://wms.geo.admin.ch/?SERVICE=WMS&VERSION=1.0.0&REQUEST= GetMap&FORMAT=image/png&LAYERS=ch.swisstopo.pixelkarte-pk25.metadata- kartenblatt&SRS=EPSG:21781&STYLES=&WIDTH=1895&HEIGHT= 1185&BBOX=475000,68000,854000,305000&SLD=http://my.server/style.sld

The WMS/SLD GetMap operation allows to reference a style authored by the user client, either hosted on an external server (parameter SLD) or directly sent with the WMS request (parameter SLD_BODY).

```xml
<?xml version="1.0" encoding="UTF-8" standalone="yes"?>
<StyledLayerDescriptor version="1.0.0"
 xmlns="http://www.opengis.net/sld"
 xmlns:ogc="http://www.opengis.net/ogc"
 xmlns:xsi="http://www.w3.org/2001/XMLSchema-instance"
 xsi:schemaLocation="http://www.opengis.net/sld
 http://schemas.opengis.net/sld/1.0.0/StyledLayerDescriptor.xsd">

<NamedLayer>
 <Name>ch.swisstopo.pixelkarte-pk25.metadata-kartenblatt</Name>
 <UserStyle>
  <Name>LabelBlattnummer</Name>
  <FeatureTypeStyle>
   <Rule>
    <PolygonSymbolizer>
     <Fill>
      <CssParameter name="fill">#FFFF00</CssParameter>
     </Fill>
     <Stroke>
      <CssParameter name="stroke">#333333</CssParameter>
      <CssParameter name="stroke-width">2</CssParameter>
     </Stroke>
    </PolygonSymbolizer>
    <TextSymbolizer>
     <Label>
      <ogc:PropertyName>Blattnummer</ogc:PropertyName>
     </Label>
     <Font>
      <CssParameter name="font-family">arial</CssParameter>
      <CssParameter name="font-size">18</CssParameter>
     </Font>
     <Fill>
      <CssParameter name="fill">#000000</CssParameter>
     </Fill>
    </TextSymbolizer>
   </Rule>
  </FeatureTypeStyle>
 </UserStyle>
</NamedLayer>
</StyledLayerDescriptor>
```

**Figure 6 Visualization of the grid of map sheets of Switzerland (1:25,000) through another cartographic facet showing labels based on the sheet number.**

In other words, the user client (e.g., Mr Tüftel) does take the control of the rendering process that may be distributed among many WMS servers. Indeed, this ability to drive remotely from the user client side (with a map viewer including a style editor) the WMS rendering server does open interesting doors to bring to life the other use cases.

## Use case "catalog"

Going further than using a simple WMS GetMap request to get a ready-to-visualize map layer, the deprecated implementation specification (version 1.0, released in 2002) of the WMS/SLD standard (*Lalonde, 2002*) does offer style management requests like GetStyles. So you get also the underlying symbology instructions of an internal style that has been predefined and used by the server to show a prepared cartographic facet of some spatial data of the underlying datasets. Thus, the retrieved style is ready to be reworked by the user client within a cartographic tool (Fig. 7). While such an ability is already interesting for the use case level 2, the SLD 1.0 style management offers not only GetStyles operation but also PutStyles operation. Together, these operations are a good start for the use case level 3 to build a catalog of styles. The WMS service is then also the storage point to discover, import and export styles to share with other SDI users through a catalog service.

Nonetheless, it is to notice that the newest SLD 1.1 release does not specify anymore the style management requests which is then a step back.

## Use case "collaborate"

Finally, for the use case level 4, the SE standard is also a centerpiece (Fig. 8). As experimented by *Bocher et al. (2012)* in the frame of the SOGVILLE/SCAPC2 research projects, SE

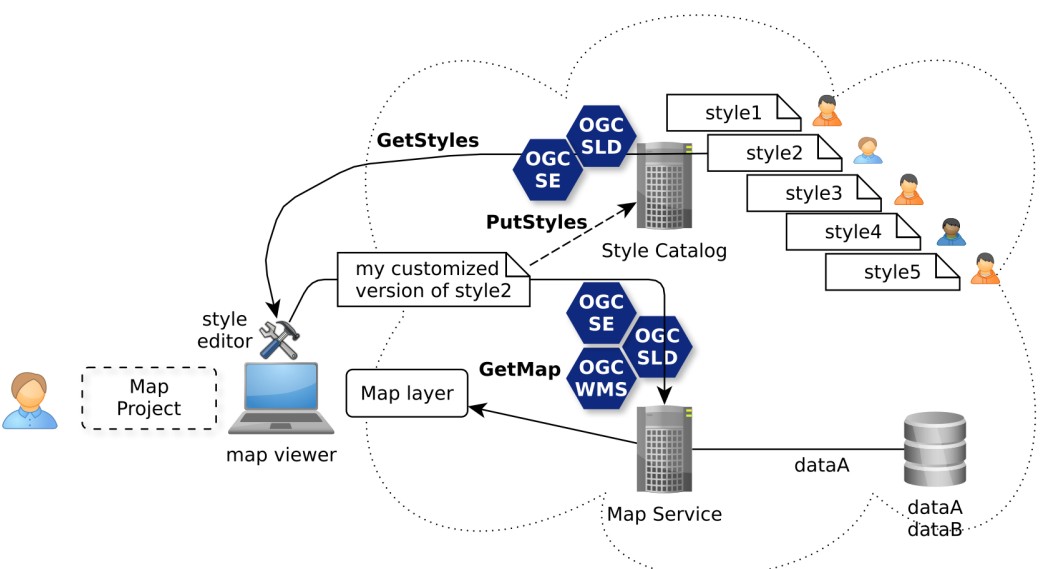

**Figure 7** Re-authoring of styles shared through catalogs with OGC WMS/SLD standards.

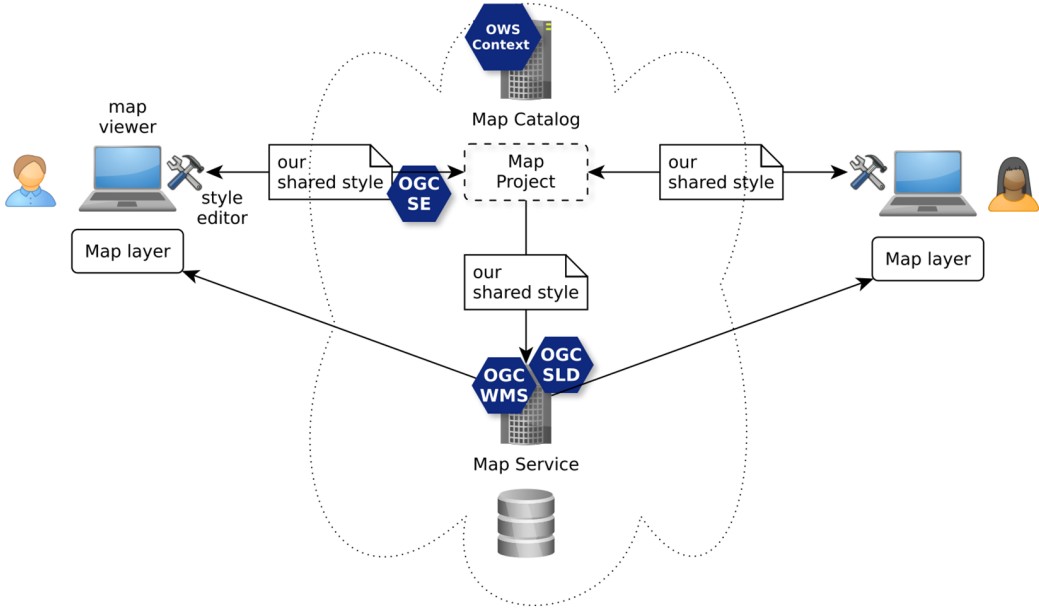

**Figure 8** Creation of a common map based on shared styles with OGC WMS/SLD, SE and OWS Context standards.

instructions are encapsulated into a structure of map project that different users share and work together in the frame of a collaborative cartographic authoring process. Indeed, while the OGC OWS Context standard is used to formalize the map project, it does in particular consider SLD and SE to formalize the shared styles used to render the map layers.

Currently, SLD (SLD 1.0) or SE (SE 1.1) (as styling language to formulate symbology instructions) are the more advanced open standards for sharing cartography as illustrated

by the above use case levels. These standards are quite largely adopted by server-side rendering systems. It can be explained because SLD is a WMS application profile which is web service oriented. Indeed, *Andrae et al. (2011)* redraws the OGC portrayal model by showing clearly SLD as the web interface to take control of the rendering engine behind the WMS service. But in 2005, the WMS/SLD 1.1 profile has been released in particular with the aim to extract the symbology instructions into a dedicated standard, the SE standard (SE 1.1). As a consequence, while the SLD profile stays strongly related to WMS service, it is no longer the case for the symbology instructions which can now be used by any styling software component, not only by WMS/SLD.

Nonetheless, at the desktop-side there are only few software which correctly and completely implement SE standard together with a graphical user interface to (re)work styles. Indeed, according to *Bocher et al. (2011)* many implementations have a conformance that is often not fully observed leading to interoperability defects in term of rendering quality. Apart from inherent bugs and dysfunctions of a tool, several reasons can explain this general situation.

- Due to a partial implementation—see MapServer implementation (*McKenna, 2011*), there are unimplemented symbology instructions, e.g., linejoin and linecap of LineSymbolizer;
- Due to the existence of two versions of symbology instructions between SLD 1.0 and SE 1.1, these tools may not check this correctly which causes parsing problems of the XML encoding;
- Due to the divergent reading of what the SE 1.1 standard tries to specify which may result in different graphical visualizations (it means there are uncomplete or ambiguous explanations in the specification—like the MarkIndex capability which doesn't specify anything on how to select an individual glyph);
- Related to the previous point, there is currently no substantial testsuite within the OGC Compliance and Interoperability Testing Initiative ("CITE") to help to disambiguate and test the graphical rendering conformance of an implementation. Beyond encoding validity and level of conformance of an implementation (range of supported capabilities), visual interpretation is essential (see Annex A in *Müller (2006)*). For instance, by comparing the output of a system to test with the output of the reference implementation.

While the above arguments do show how it is essential to have a common styling language (currently in the name of OGC SE 1.1), this importance is accentuated by the fact that many changes and proposals have been received by the standard working group (SWG), in particular from the scientific community (*Duarte Teixeira, De Melo Cuba & Mizuta Weiss, 2005*; *Cooper, Sykora & Hurni, 2005*; *Sykora et al., 2007*; *Dietze & Zipf, 2007*; *Sae-Tang & Ertz, 2007*; *Schnabel & Hurni, 2007*; *Mays, 2012*; *Iosifescu-Enescu, Hugentobler & Hurni, 2010*; *Bocher et al., 2011*; *Rita, Borbinha & Martins, 2012*; *Bocher & Ertz, 2015*). All these works share a common claim about enhancing SE. It seems the communities of users were frustrated because no substantial new symbology capabilities

have been introduced with the release of SE 1.1 except transformations functions. Moreover, *Bocher et al. (2011)* and *Bocher & Ertz (2015)*, explain that these only new and few capabilities (interpolate, recode, categorize functions) cause confusions and even some regressions.

For instance, despite all the good intentions, there are several limits that come out from the introduction of the categorize function (defined by SE 1.1 standard as the transformation of continuous values to distinct values, e.g., useful to build choropleth maps):

- The definition seems to only match a requirement emphasized by Jenks (*Slocum et al., 2009*) that classes must cover all the possible values of the dataset and must not be discontinuous. However, such a definition has limits considering optimal methods like the Jenks–Fisher classification or Maximum Breaks classifications that may produce intervals with gaps (*Slocum et al., 2009*) and that it is often better to use the lowest value of the dataset as the minimum value of the first interval rather than negative infinity;
- The categorize function is redundant with the concept of Rule of the SE standard. Moreover, the latter does offer wider possibilities to define precisely value intervals (minimum/maximum values instead of negative/positive infinite, non-contiguous intervals, interval as singleton);
- Similarly, the RasterSymbolizer concept used to control the styling of raster data has been reduced because of the ColorMapEntry concept from SLD 1.0 has been replaced by the categorize transformation function;
- Finally, the introduction of categorize function has also removed from SLD 1.0 the capability to associate a label to an interval when it is an important requirement to have such an information to build a map legend.

Along the same lines, the many proposed extensions of SLD and SE standards have to be analyzed. The purpose is to identify how these cartographic enhancements are relevant for the redesign of the SE standard. By way of other examples, *Sae-Tang & Ertz (2007)* describe four new possibilities to generate thematic maps (CategoryThematicSymbolizer, SimpleThematicSymbolizer, MultiThematicSymbolizer, ChartThematicSymbolizer). A similar approach appears in *Dietze & Zipf (2007)* (DiagramSymbolizer and ChoroplethSymbolizer) and in *Iosifescu-Enescu, Hugentobler & Hurni (2010)* to support various diagram types (e.g., pie charts, bar diagrams) to fulfill the complex visualization requirements coming from environmental management.

Also, the specific options introduced within the XSD schemas by some off-the-shelf geospatial software (e.g., "GeoServer") have to be considered. Of course the extensible nature of XML is convenient to add cartographic capabilities to implement in the software, but it may at the same time also create some non-interoperable defects. Clearly, it seems SE 1.1 has never been designed with modularization and extensibility in mind and there are no explicit extension points defined in the underlying symbology model. Moreover, the SE standard does currently only offer one XML-based encoding and strongly linked to XML modeling principles (Fig. 9). As a consequence, it may be

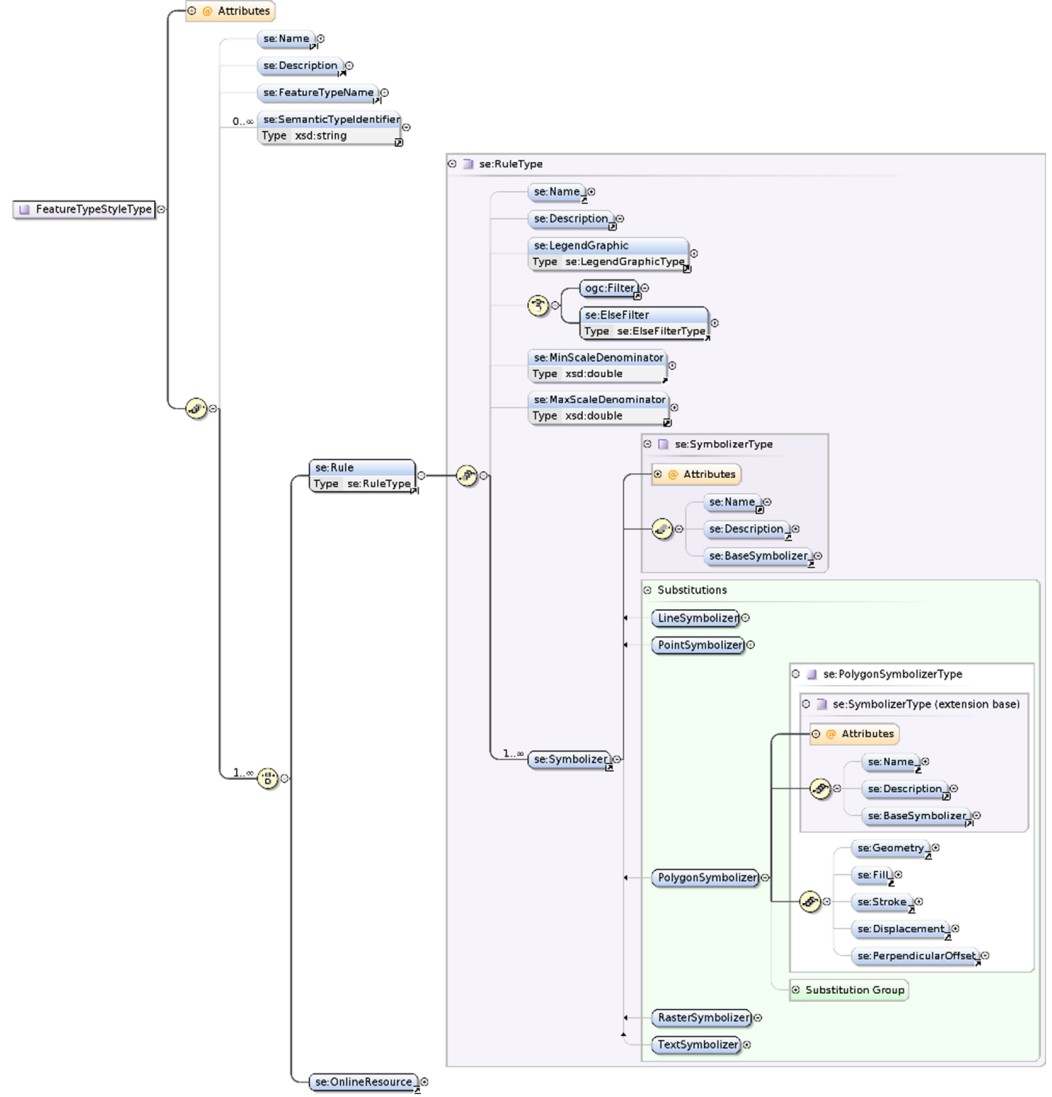

**Figure 9 The physical symbology model of SE formalized with XML Schema Definition.**

difficult for cartographic communities and developers having different encoding preferences (e.g., CSS-like or JSON-based) to get a chance to observe conformance. Indeed, while there is a general trend to dislike XML, other encodings seem to be in fashion, like the YAML-based YSLD styling language proposed by *GeoServer (2017)* in addition to the support of OGC SLD standard, or the CSS-derived styling languages MapCSS (*OpenStreetMap, 2017*) or CartoCSS styling language from *Mapbox (2017)*, although it seems already old-fashioned (*MacWright, 2016*). Also, there are major proponents of an encoding which would make a wider use of relevant and famous graphical standards like SVG, just like OWS Context does use the famous Atom syndication format (*Brackin & Gonçalves, 2014*). Beyond the trends, there is no consensus by now.

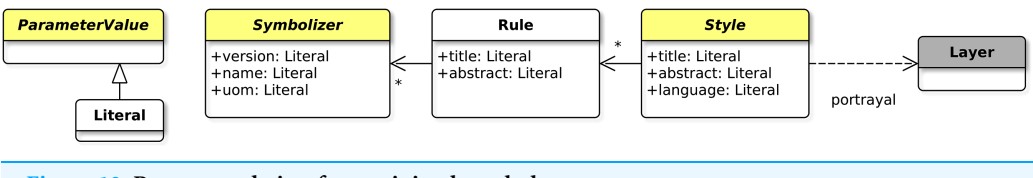

**Figure 10 Recommendation for a minimal symbology core.**

To conclude this chapter, while there are clear possibilities to implement the four levels of sharing cartography, it is also clear that a revision of the common styling language played by the SE standard is required. Three major requirements have to be considered:

- Enrich the standard with new cartographic capabilities inline with the evolution of the needs coming from the map-makers community;
- Redesign the underlying symbology model of the standard so as to be modular and extensible for the long-term;
- Consider the possibility to have other encodings than XML.

The next chapter does develop some proposals to fulfill these requirements.

## PROPOSALS

The overall purpose is to make standards dedicated to cartography (in particular SE) more attractive by turning them into "a really useful (cartographic) engine," quoting the nod to Thomas the Tank Engine alluded by the OGC "Specification Model—A Standard for Modular specifications" document (*Policy SWG, 2009*), called the modular spec in below.

Before compiling all the Change Requests collected by the SLD/SE SWG, one question does arise: how to plug a new requested ability in the standard? One first and fundamental recommendation is then to consider the modular spec whose release 1.0 has been edited in 2009, at the time the SE standard was already released and thus not in compliance with. Indeed, the modular spec specifies generic rules to organize the internal logical structure of the standard in a modular way so as to strengthen the guarantee of a useful and worth standard easy to implement but also to extend.

### Modular structure: one symbology core, many symbology extensions

The modular spec fittingly suggests modularity with the idea of a standard built of one simple core and many extensions which expand the functionality of the specification. Applied to a new revision of the SE standard, the definition of a symbology core requires first to "reverse design" the underlying symbology model of SE 1.1. After which, the concrete symbology capabilities have to be extracted and split into many relevant extensions while taking care of dependencies. The proposed minimal symbology core illustrated by Fig. 10 is partially abstract and defined according to the following concepts:

- The Style concept, in charge of the cartographic portrayal of a collection of features stored within a Layer by applying at least one symbology Rule. A feature is described as an abstraction of real world phenomena as defined by GML standard (*Portele, 2007*);

- The rendering does run feature per feature using a "one drawing pass" engine;
- Each Rule may be scale filtered and does hold at least one Symbolizer;
- Each Symbolizer does describe the graphical parameters for drawing the features (visual variables);
- The Style, Rule and Symbolizer concepts hold parameters which are literal values.

Some of the concepts are defined as abstract (in yellow and with italic names in Fig. 10) so as to be considered as extension points. Actually, regarding this, we may notice that *Craig (2009)* does request a similar concept by the use of XML abstract elements which may than be considered as extension points.

Now that the core is ready, some surrounding extensions may be defined so that the engine is really able to perform a rendering. Indeed, alone, the core does not concretely "do" anything. As an example, let us introduce the AreaSymbolizer extension which holds a simple and classical symbolizer, call it the AreaSymbolizer concept which describes the graphical parameters for drawing polygonal features with outlined and filled surface areas. The aim of the below explanations is to illustrate with a simple example the extension mechanism and how extension points are expanded.

At first, it is defined that the AreaSymbolizer extension has a dependency with the FeatureTypeStyle extension and the related concepts:

- The FeatureTypeStyle specialization of the Style core concept;
- The portrayal of a Layer built of *N* instances of GML AbstractFeatureType (*Portele, 2007*);
- The ability to access features according to Simple Feature SF-2 (*Van den Brink, Portele & Vretanos, 2012*);
- The geometry parameter to each Symbolizer extension that depends on this extension (in this case the AreaSymbolizer extension).

Then, given that the geometry parameter is defined with a dependency on the ValueReference extension, the ValueReference specialization of the ParameterValue core concept is introduced. In a general way, when a parameter has to be assigned with a value, ValueReference does introduce the ability to reference the value extracted from a data attribute of a feature. This is useful when a FeatureType does hold many geometry properties and allows to reference the one to be used by the renderer.

Finally, the AreaSymbolizer extension itself is required, holding the AreaSymbolizer specialization of the Symbolizer core concept. Called PolygonSymbolizer in SE 1.1 and correctly renamed AreaSymbolizer by *Craig (2009)*, it does introduce:

- The symbology ability to draw a surface area according to a filling and an outline;
- The dependency on the FeatureTypeStyle, Fill and Stroke extensions;
- The ability to reference the geometry data attribute to be drawn (by means of its dependency on the FeatureTypeStyle extension).

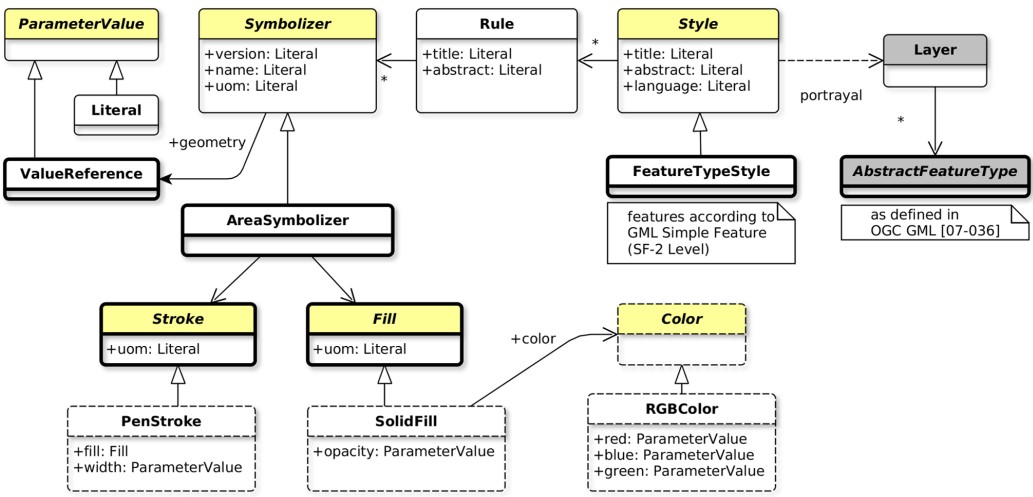

**Figure 11 Concepts to implement so as to observe conformance with the AreaSymbolizer extension.**

In consequence, an implementation that wants to observe conformance with the AreaSymbolizer extension requires to implement and drive its rendering engine according to all the concepts of the core (thin outline in Fig. 11) and the AreaSymbolizer concept with all the other concepts required by dependencies (bold outline in Fig. 11).

Nonetheless, even at this point, a rendering engine would neither concretely "do" anything. Indeed, the implementation has then to offer choices related to the filling and the outline. Some more concrete capabilities have to be implemented, for instance with (dashed outline in Fig. 11):

- The SolidFill concept, a Fill specialization which introduces the graphical ability to define a solid color value combined with an opacity;
- The PenStroke concept, a Stroke specialization which introduces the graphical ability to draw a continuous or dashed line with or without join and cap;
- The dependent abstract Color concept (and again a concrete choice of color definition has to be done, like with the RGBColor concept which defines a color in the sRGB color space with three integer values).

Having this modularity approach for long term extensibility applied to all the symbolizer concepts, past, present and future, an implementation can with ease manage step by step the evolution of the conformance level of its technical implementation of the standard.

## One encoding-neutral conceptual model, many encodings

Currently, SE 1.1 offers a physical model using XML Schema Definition and, at the same time, a natural encoding based on XML. The initial motivation explaining the below recommendation is related to the fact that there is not only XML, but also many other flavors of encoding, JSON-like, CSS-like, YAML-like among many others it is possible to imagine. The important for portrayal interoperability is not the encoding, it is rather the

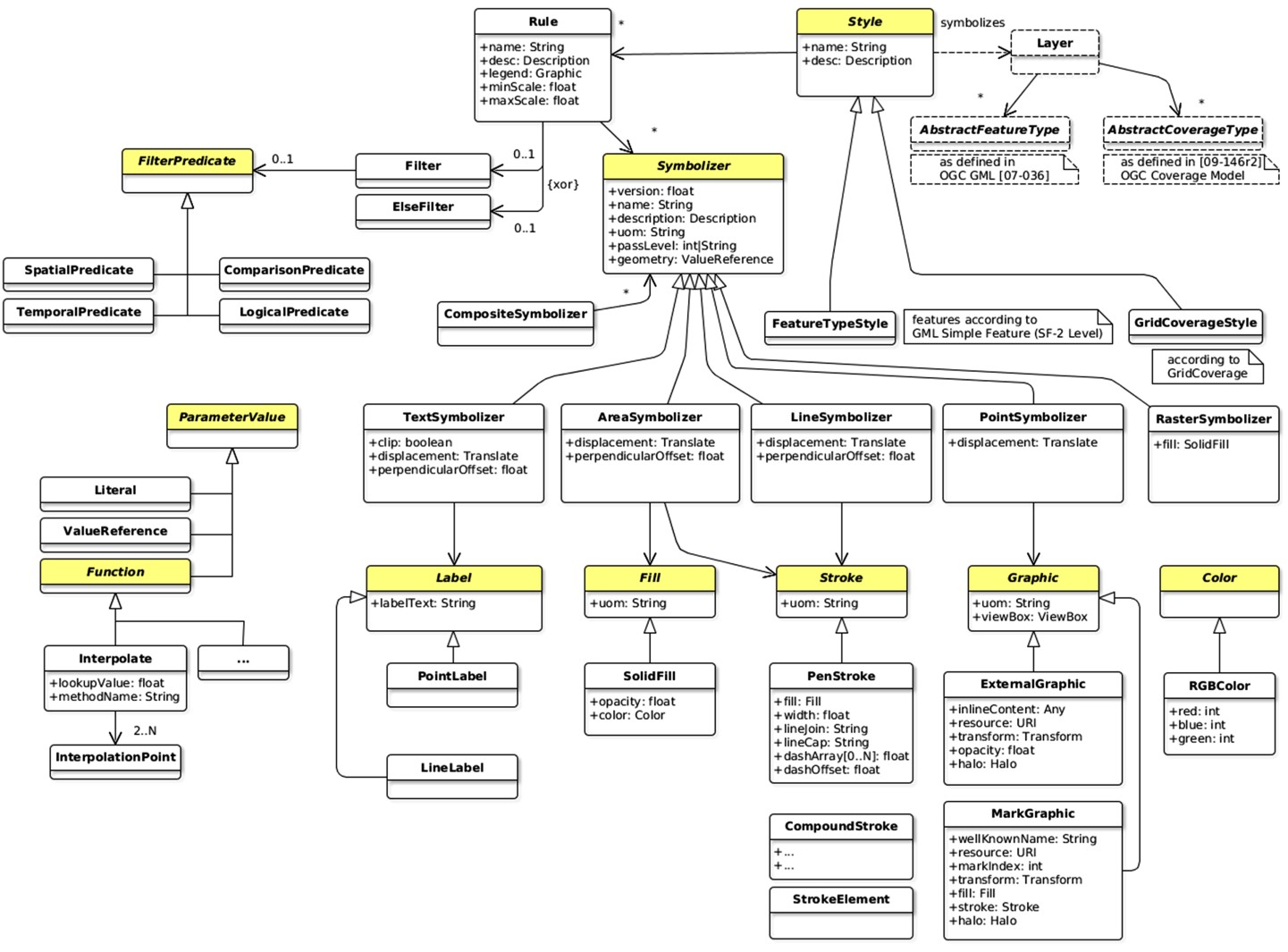

**Figure 12 Extract of the proposed symbology model.**            

symbology model. That is why the "one encoding-neutral model/many encodings" approach is promising to favor a large adoption of the standard.

This approach has on one side the encoding-neutral model formalized using UML notations, it can be considered as conceptual. With a class diagram, it does describe the portrayal concepts, their relationships, the modular organization, the extension points and the dependencies. We may notice that UML is often preferred when some work is about the design of portrayal concepts. In *Zipf (2005)*, a simplified version of the underlying symbology model of SE 1.1 is depicted as an UML class diagram. Moreover, *Craig (2009)* does suggest to avoid the XSD attribute concept in the XML encoding so as to be more portable to other structuring languages which do not have the unusual attribute concept of XML Schema, UML in particular. These are more arguments that are in favor of defining at first a conceptual and encoding-neutral model (Fig. 12).

Consequently, doors are open to offer a variety of encodings. Each encoding does translate into a format the UML notations according to mapping rules. At least one

default encoding and following the OGC tradition, XML may be this default encoding. It is up to the SWG to define the mapping rules to translate the semantic of the conceptual model into XML Schema definitions. Indeed, as noticed by *Lonjon, Thomasson & Maesano (2006)*, the translation from UML to XML requires a thoughtful analysis of the conceptual model so as to define the global mapping rules (e.g., translate a specialization relationship using static or dynamic typing? how to translate a concrete class, an abstract class, the various types of associations? when using attributes or elements?, etc.). Thus, UML and XML are together a winning combination two times inline with the modular specification which recommend UML "If the organizing mechanism for the data model used in the specification is an object model" and XML "for any specification which has as one of its purposes the introduction of a new XML schema."

Of course, all these questions related to the mapping rules have to be considered for each encoding offered with the standard. We may notice that the OWS Context SWG adopted a similar approach, offering the default encoding based on XML Atom and planning to provide an OWS Context JSON Encoding soon, according to *Brackin & Gonçalves (2014)*.

## Style management and parametrized symbolizer

Beyond the tempting recommendation to reintroduce the WMS/SLD GetStyles and PutStyles methods, the management of a catalog of styles has to be expanded. Thus, *Craig (2009)* does suggest the introduction of a mechanism to reference the definition of a Symbolizer hosted within a catalog. Moreover, the report does enrich the referencing with a symbolizer-parameterization mechanism so as to offer complete symbolizer re-usability between different, incompatible feature types. It consists of a list of formal-parameter names and an argument list.

It is to notice that such a mechanism does fit the one specified by *ISO (2012)* in term of parameterized symbol built of dynamic parameters. Thus, in a general way, it is recommended to consider what ISO has already specified concerning the concepts of "collection of symbols and portrayal functions into portrayal catalog."

Concerning this aspect of style management, the proposal suggests to continue the conceptual work by blending together all these recommendations: reintroduce GetStyles/PutStyles and introduce the mechanism of symbolizer-parameterization inline with *ISO (2012)*.

## New symbolization capabilities

Among the many symbology capabilities that can be extracted from the pending Change Requests at OGC and the research works, we list below (non exhaustively) some relevant ones. Considering the modular structure (see A), each of these capabilities is an extension (e.g., HatchFill is an extension of the Fill abstract concept, just as SolidFill):

- UnitOfMeasure: current SE 1.1 standard does only offer two ground units (meter and foot) and one portrayal unit (pixel, which is also not an absolute unit of measure). It may be relevant to add at least three additional units to make measurements more

portable between styling representations and rendering environments: portrayal millimeters and inches as printing measurements, and portrayal (printer's) points commonly used for font sizes;

- Transformations: currently, SE 1.1 standard does offer only locally few transformations capabilities (translation of a polygon or graphic, rotation of a graphic). It may be relevant to spread out all kind of general affine transformations like Translate, Rotate, Scale, Matrix using homogeneous coordinates on geometries and graphics;

- Functions: currently, SE 1.1 standard does extend the concept of ogc:expression inherited from the deprecated Filter Encoding 1.1 standard (*Vretanos, 2001*) to adequately support the needs of symbolization in transforming (categorization, recoding, and interpolation) and editing data (formatting numbers, strings and dates). It may be relevant to directly use the function definition mechanism of Filter Encoding 2.0 standard (*Vretanos, 2010*) rather re-inventing such a mechanism (*Craig, 2009*);

- CompoundStroke: current SE 1.1 standard does offer simple stroke just like with a pen (optionally with dash pattern) or the linear repetition of a graphic. It may be relevant to allow multiple graphic and/or simpler strokes to be combined together along the linear path. It is interesting to produce complex stroke styles such as rendering a sequence of graphic icons along a line or drawing simple dashed lines between boat-anchor icons (*Craig, 2009*);

- CompositeSymbolizer: currently, grouping of symbolizers is only possible in relation with a rule, eventually driven by a filter. It may be relevant to manage descendant symbolizers as a single unit separately from the definition of a rule. Having a dedicated concept for grouping symbolizers does make the logical grouping more explicit and allows a group of symbolizers to be remotely referenced (see the SymbolizerReference concept in *Craig (2009)*);

- HatchFill: currently, SE 1.1 standard allows one color filling and the repetition of a graphic to fill an area. It may be relevant to add cross hatching, a method of area filling which is often used and has so simple parameters that it should be established as another filling variety. It is required to allow the configuration of such a filling in a way conventional in cartography, otherwise the user would be forced to emulate cross hatching by fiddling with the GraphicFill concept;

- DiagramSymbolizer: current SE 1.1 standard does allow the use of graphics generated externally (e.g., static image) or well-known shapes or font glyph whose color can be set internally. It may be relevant to allow the internal definition of more complex diagram symbolization of geographic features like "Pie," "Bar," "Line," "Area," "Ring," and "Polar" charts. Indeed, it is a usual and effective way of visualizing statistical data (*Iosifescu-Enescu, 2007*);

- Multiple drawing pass: current SE 1.1 standard does describe a one drawing pass rendering (driven by applying symbolizers in the order they are defined by the style and according to rules and filters). It may be relevant to better control the rendering with the capabilities to order the level of symbol rendering (e.g., to draw nicely connected highway symbols).

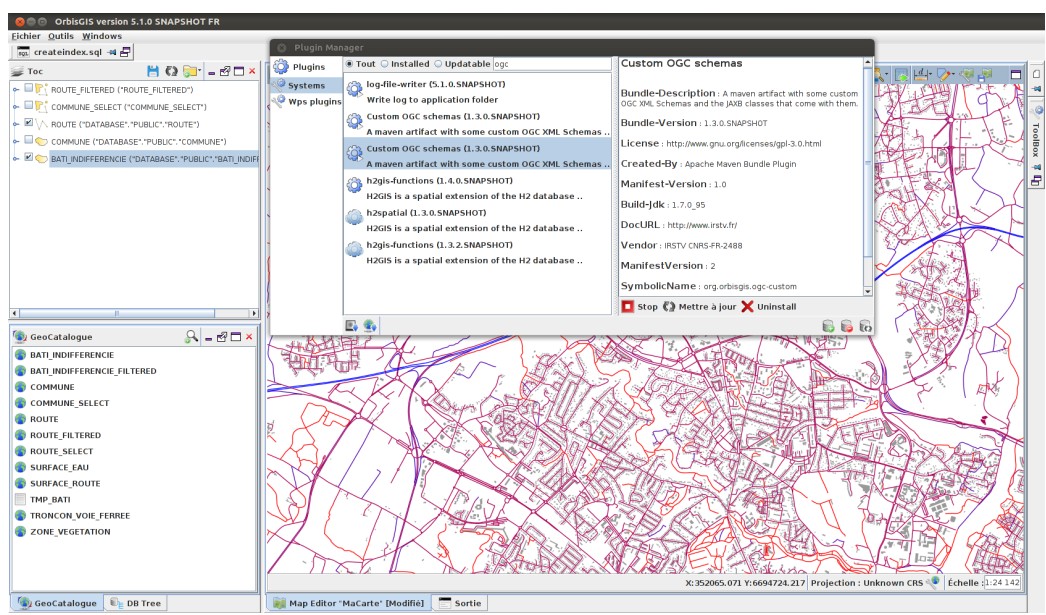

**Figure 13** OrbisGIS dynamic module system with OSGi.

## REFERENCE IMPLEMENTATION

The OrbisGIS platform has been used to prototype an implementation of the symbology model all along the standardization work by iterations with tests and validations (*Bocher & Petit, 2015*). In the long term, this platform might be adopted as a reference implementation at the OGC ("CITE").

OrbisGIS is a Geographical Information System designed by and for research (*Bocher & Petit, 2013*) which is the main advantage for research communities comparing to other GIS. Indeed, OrbisGIS does not intend to reproduce classical GIS functionalities. It is designed to explore new issues or questions in the field of geospatial techniques and methods (such as language issues to query spatial information and issues on cartography about standardization, semantics and user interface design). To address these challenges, the OrbisGIS architecture (object and data model) and its user interface are frequently redesigned. This approach is fundamental to test the concepts and the ideas related to the ongoing standardization process of symbology standards at OGC. Furthermore, the fact that we have a common set of GIS features organized with the dynamic module system OSGi to access to the geodata, library to use simple features functions, layer model, rendering engine, etc. (*OSGi, 2014*), gives flexibility to plug some experimental code without breaking the platform and the user can easily switch from one to another plugin (Fig. 13). More importantly, the usage of OSGi technology does offer a way to implement the modularization principles depicted in the above (i.e., one OSGi bundle per symbology extension).

Another motivation is related to the license. OrbisGIS is an open source software, distributed under the GPL3 license and therefore grants four freedoms (1) to run the program for any purpose, (2) to study how the program works and adapt it to your needs,

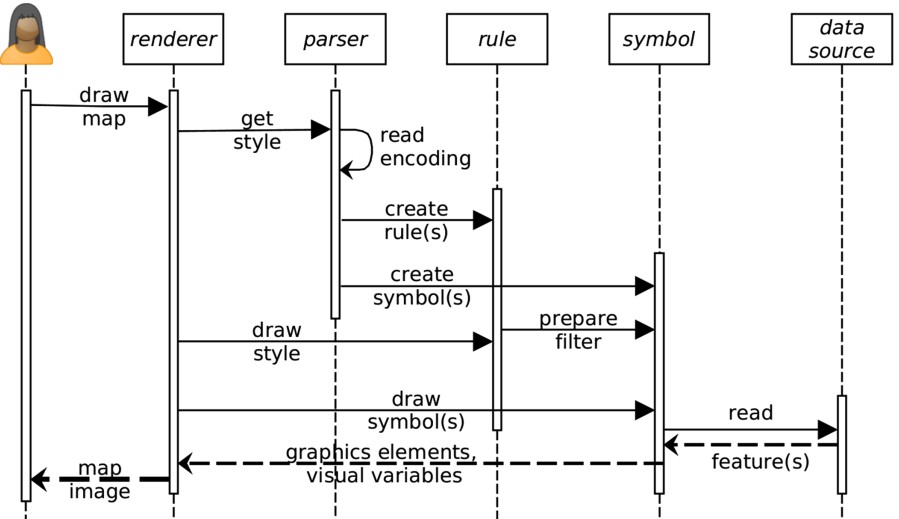

**Figure 14  Main sequences of the rendering engine.** 

(3) to redistribute copies so you can help your neighbour, and (4) to improve the program, and to release your improvements to the public, so that the whole community benefits (*Steiniger & Hunter, 2012*).

This aspect is essential in order to have a reference implementation available for the community of implementers of a standard, guiding them better in the understanding of a specification. Given the core principle of science that having open source code available does enable reproducibility (*Ertz, Rey & Joost, 2014*), we argue that this is also valid for open standards. On one side, it is easy for other researchers and businesses to verify and re-use new developments and adapt them to their needs (*Steiniger & Hunter, 2012*). Furthermore, having the code of the rendering engine, the user interfaces and all the tests fully accessible should facilitate the understanding and the dissemination of standards for portrayal interoperability while minimizing interoperability defects. In the following we describe the main aspects covered by OrbisGIS to implement the proposed redesign of the symbology model.

## XML encoding/decoding

In the context of a prototyping iteration, the symbology model presented in the chapter 4 has been transposed to a XSD schema (*Maxence et al., 2017*). The Java Architecture for XML Binding (*Ort & Mehta, 2003*) library is used to generate the XSD schema-derived Java binding classes. Finally, a Java Style Object Model is built. Thus, symbology instructions are stored in a style file using XML encoding and is parsed prior to be applied by the rendering engine.

## Rendering engine

The rendering engine is a OSGi bundle whose mechanism is divided into 12 sequences (Fig. 14):

(1) User interface event to draw a map.

(2) The renderer engine gets the style file that contains the symbology instructions.

(3, 4 and 5)  The style file is read by the XML parser to create the Java Style Object Model composed of rules and symbols.

(6)  The renderer engine starts to draw the style object looping over each rules.

(7)  Each rule is scanned to check if a filter must be applied. The filter condition (e.g., select all values greater than . . . ) is prepared for each symbolizer of the rule.

(8)  The renderer engine starts to draw all symbols available in the Java Style Object Model.

(9)  Each symbol reads the data source on which the style must be applied.

(10)  A set of features according to the potential filter constraint of the symbolizer is returned (including geometries and data attributes).

(11)  The symbols are filled with the features properties to create the graphic elements and visual variables.

(12)  Finally, the renderer engine displays the style as a map image.

## User interfaces

OrbisGIS offers two kind of user interfaces for configuring the map styles using the capabilities of the underlying symbology model (Fig. 15):

- At first some productivity tools organized around a set of widgets each dedicated to common thematic maps. The possibilities are limited to what these widgets are able to configure related to what they have been built for. Nonetheless, the second tool can then be used in an expert mode to go further.

- Secondly, rather intended for an expert who want to tinker and tweak. As an advanced style editor, it is a flexibility tool which allows to manipulate all elements of the symbology model (Rule, Symbols, visual variables). A good knowledge of the symbology model is required because each elements of the style must be set individually. Consequently, the user can express without any limitation (except the limits of the symbology model itself) all her(his) creativity to build cartographic visualizations.

To illustrate some results rendered with OrbisGIS we present two maps extracted from the "Wall of Maps" (*Bocher & Ertz, 2016*). The first one shows a bivariate map to display the number of building permits in Europe in 2005 compared to 2014 (Fig. 16).

Bivariate map is a common technique to combine visual variables. The map uses the same type of visual variable to represent two values (as half circles). The main symbology elements used to create this bivariate map are:

- The style element contains two rules named A and B;

- Rule A contains one symbolizer element (AreaSymbolizer) to display the stroke of the European countries;

- Rule B defines the bivariate proportional symbol with two elements of PointSymbolizer (for readability, we present only the instructions for the left half-circle visual variable);

- The PointSymbolizer contains several sub-elements:

  – The geometry element allows specifying which geometry attribute is to be rendered;
  – The ST_PointOnSurface is an OGC filter function (*Vretanos, 2010*) used to have a point geometry guaranteed to lie on the surface. This new point derived from the input geometry is the location where to anchor a MarkGraphic, otherwise the symbol might be applied on all the vertices of a geometry;

- the MarkGraphic is defined by:

  – The symbol shape identified by a well-known name, HALFCIRCLE (right side);
  – The size of the shape varies according the height of its view box;
  – To have the shape size proportional with the number of building permits in 2015:

    ∗ An interpolate function is applied on;
    ∗ It uses a ValueReference that points to the attribute named permits2005;
    ∗ The interpolation is defined by two interpolation points chosen along a desired mapping curve (here the minimum and maximum values);
    ∗ For each interpolation point the height of the view box is specified with a specific unit of measure;

  – Because the half-circle shape is drawn to the right side, a 180° rotation is operated;
  – To finish, the MarkGraphic is filled with a RGB color.

The second map shows a combination of several visual variables: shape, size, color, patterns and orientation (Fig. 17). The style is organized around six filtered rules that correspond to the biogeographic regions in Switzerland. We present two Rules (A and B) that use the HatchFill and GraphicFill concepts which are extensions of the Fill abstract concept of the symbolizer model.

## CONCLUSION

Considering the fundamental works of *Bertin & Berg (2010)* and successors, the community of map makers has constantly investigated questions about cartographic visualizations in term of design using the appropriate visual variables and combining them together with relevancy. Despite an important body of principles and practices, the community did not grasp the questions about standardization. However, given the multiplicity of software used to flood the world with maps, these questions are nowadays a strategic challenge to be considered in relation with operational requirements.

Even if the definition of a cartographic-oriented standard is not able to act as a complete cartographic design framework by itself, we argue that pushing forward the work aiming at the creation of dedicated standards for cartography is a way to share and disseminate good practices. Indeed, too much SDIs do merely accept the limits of the current standards and consequently poor map design and quality. While they have to apply OGC standards, it is essential to build standards so as to be able to enrich their

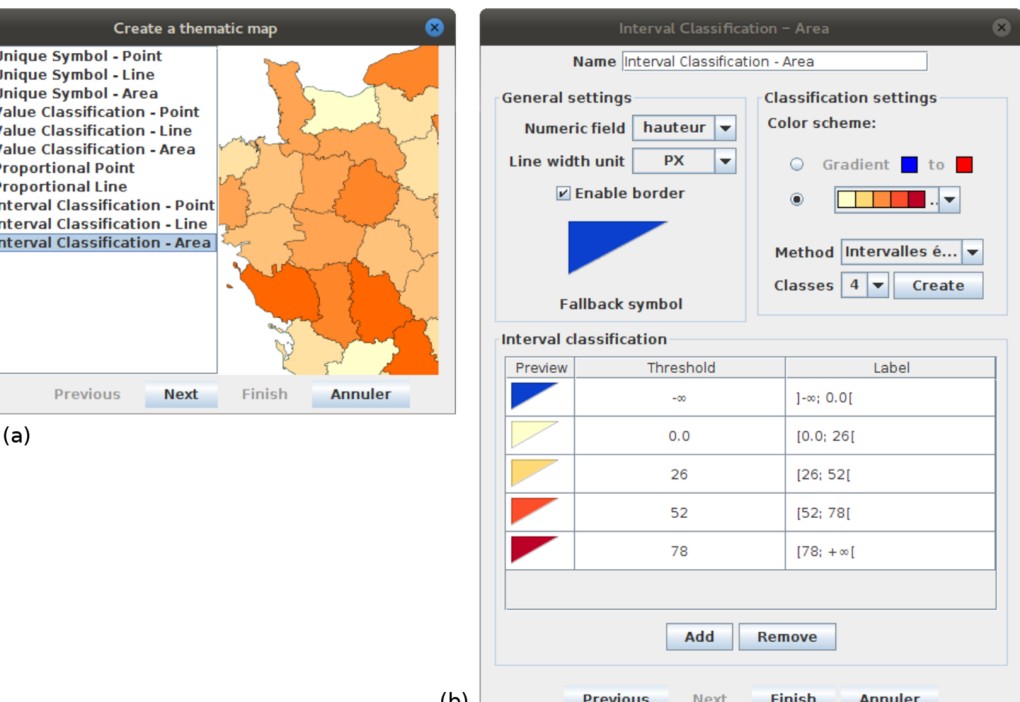

(a)

(b)

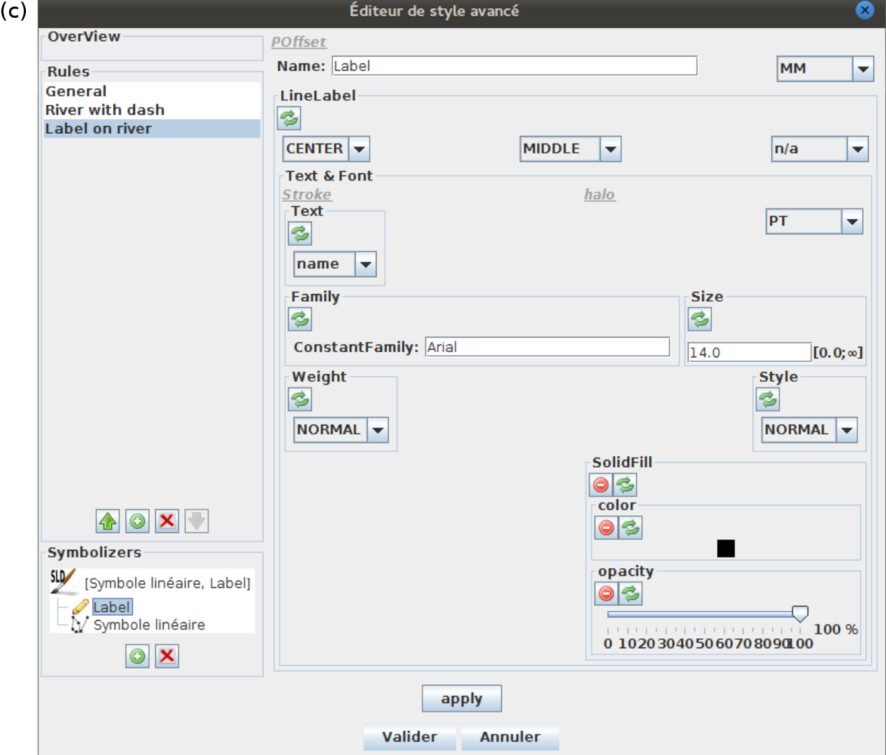

(c)

**Figure 15** (A) The screenshot shows the list of productivity tools available in OrbisGIS. (B) The screenshot shows the user interface of the productivity tool dedicated to choropleth maps. (C) The screenshot shows a prototype of advanced style editor.

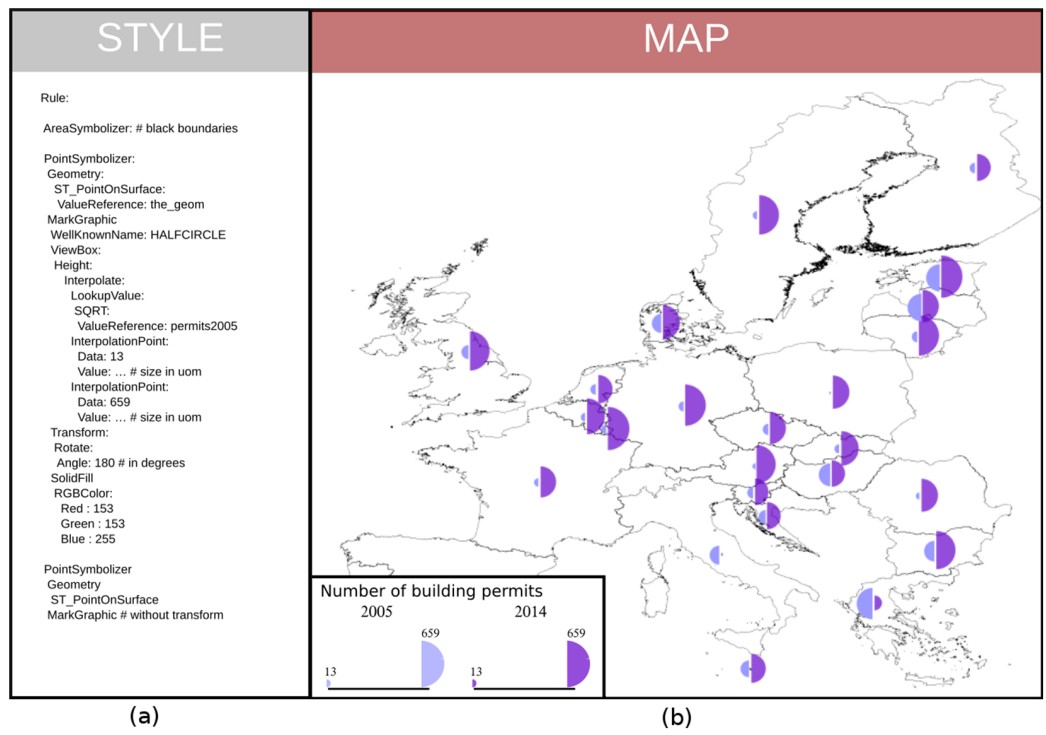

**Figure 16** (A) Some redesigned symbology instructions (YAML encoded for the ease of reading). (B) A bivariate proportional symbol map coming out from the rendering engine using these instructions.

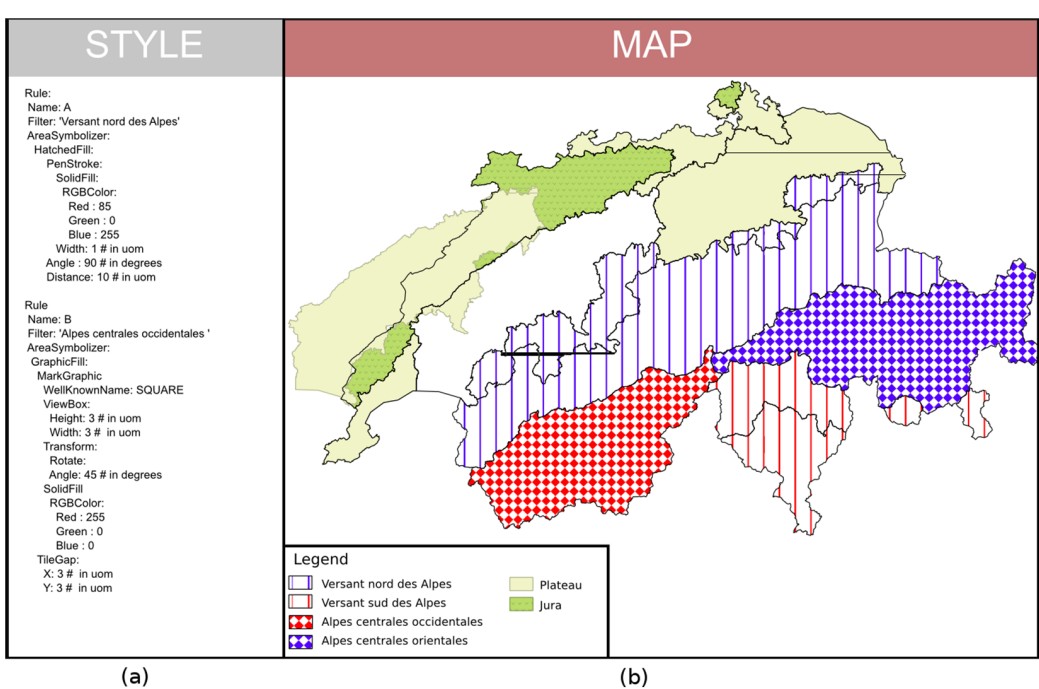

**Figure 17** (A) Symbology instructions showing how to combine visual variables (YAML encoded for the ease of reading). (B) A map of the biogeographic regions in Switzerland coming out from the rendering engine using these instructions.

cartographic capabilities at long-term, to make grow up the good practices and finally to improve the quality of the visualizations. In this sense, we have identified some use cases showing how it is important to make portrayal interoperability operational for sharing cartography, from discovery to collaboration activities, by way of authoring and cataloging activities.

From research results in link with the dedicated SLD/SE OGC SWG (*Ertz & Bocher, 2010*), this paper does extract some recommendations to enable portrayal interoperability. They invite to improve the OGC SE standard based on principles and practices in cartography. We start from a functional definition of a map translated into a set of visual variables which are combined to create symbols and finally a map style. The proposed recommendations do observe this functional definition which is already at the heart of how SE standard has been specified by OGC.

Now, in the long term, it is recommended that a design approach is driven by a conceptual definition of the model and unconstrained by specific encoding aspects, and, as soon as the model is ready, then a default encoding is offered (e.g., XSD/XML). Following from this approach of dissociation, it does allow the definition of other encodings according to the various flavors within the communities.

Given that the cartographic requirements will progress over time due to practices growing up and according to domain specific features, the offered symbology model is empowered so as to be extensible and ready to offer new cartographic methods. Moreover, such a modular approach allows implementations to be compliant step-by-step. As a consequence the adoption of the standard should be favored.

Finally, we claim to a testsuite within the OGC CITE so as to help to disambiguate and test the visual conformance of the implementations. While it shall be associated to reference implementations, having at least one open source is also essential for the community of implementers, guiding them even more in the understanding of the standard. In this sense, OrbisGIS is an open source platform that has been used to prototype an implementation of the symbology model all along the standardization process by iterations with tests and validations. It might become an open source reference implementation.

### Funding
The authors received no funding for this work.

### Competing Interests
The authors declare that they have no competing interests.

### Author Contributions
- Erwan Bocher conceived and designed the experiments, performed the experiments, analyzed the data, contributed reagents/materials/analysis tools, wrote the paper, prepared figures and/or tables, performed the computation work, reviewed drafts of the paper.

- Olivier Ertz conceived and designed the experiments, performed the experiments, analyzed the data, contributed reagents/materials/analysis tools, wrote the paper, prepared figures and/or tables, performed the computation work, reviewed drafts of the paper.

### Data Availability

GitHub repository with the OrbisGIS platform, containing a partial implementation of the Symbology Rendering model: https://github.com/orbisgis/orbisgis.

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
