# Peer review of "A redesign of OGC Symbology Encoding standard for sharing cartography"

_PeerJ Computer Science, doi:10.7717/peerj-cs.143_

## Round 0.1 · original submission · Minor Revisions

Dear Olivier, you can find here below the reviewers' comments; please carefully follow the reviewers' recommendations in revising your article. Kind regards Gabriella Pasi

Reviewer 1 ·

Basic reporting

The paper needs a native English language revision to make it clearer.
It is well organized and this helps understanding the main contents.

The cited literature references related with OGC standards and portrayal interoperability of such standards are relevant.
Figures and schema help understanding the main contents and well exemplify the potentiality of the proposal.

Experimental design

The faced problem to allow on the user side of an SDI personalizing the style of visualized maps according to user cartographic preferences is original and relevant.
Also the proposal to enable such a feature by fostering the compliance with OGC standards for interoperability of SDIs as far as portrayal interoperability is a correct way to approach the problem.
Finally the implementation architecture of the proposal within the OrbisGIS is well schematized and the example shown provide an idea of the potentiality of the approach.

Validity of the findings

The validity of the proposal for personalization of SDI and the novelty of the results could be exploited for designing geoportals customized to user categories.

Additional comments

The faced problem to allow on the user side of an SDI personalizing the style of visualized maps according to user cartographic preferences is original and relevant.
Also the proposal to enable such a feature by fostering the compliance with OGC standards for interoperability of SDIs as far as portrayal interoperability is a correct way to approach the problem.
The proposal should make clearer the difference between choice of the styles of a thematic maps, with respect to choice of the graphic representation of a thematic map, for example using coremi.
Finally the authors needs to explicit how far the proposal can go in the personalization of the cartographic style, for example if anamorphic maps could be a possible choice of the style of a map.

The paper needs a native English speaker revision.

·

Basic reporting

In the introduction, your discussion on support of visualization in current SDIs is a good point, and relevant is the consideration of INSPIRE, but your conclusion that portrayal is considered a concern of second zone could sound too strong, given for instance the “data specification on geology theme” (INSPIRE D2.8.11.4_v3.0 - http://inspire.ec.europa.eu/id/document/tg/ge), where mature styling practices from the geological community are exhaustively considered and included in the guideline document (pages 106-130).
Figures are well designed and of great support to the reader. I have just some points about Figure 7 and related text (“use case catalog” paragraph, lines 230-242). In the figure you depict the “catalog” and the “map service” as two separate components, and I agree that this is a good interpretation of functionalities, but what you describe in the text concerns a single endpoint, the actual WMS implementation (“the storage point to discover, …” line 239), and this is a bit confusing with respect to the figure, where the arrow from “my customized version of style2” and the “map service” is hiding the put operation which the reader would expect towards the style catalog, while now he could interpret the current arrow having the same meaning of corresponding arrow in Figure 5 (getMap request with posted user style). Hoping to catch the overall idea, I would suggest you to solve this issue stressing the separation of functional modules. If you agree, you could slightly modify both the figure and the text: 1. in figure 7 I would add a second arrow from “my customized version of style2” directed towards “style catalog”. 2. the text could suggest the following lifecycle of the style: the user gets style2 from catalog, then he makes modifications to the style with his software (style editor) and sends a getMap request to WMS with the the customized style, repeating modifications to the style till satisfactory results are obtained (that is: as you suggested in the use case, the user does not want to download data, so he uses WMS capabilities of interpreting style), and at that point the style is put into the catalog.
A second very minor suggestion for the same figure is then to add the label “style editor” just like in Figure 5.
Minor issues
Line 265: something is missing, at least an hyphen, but also “mapServer” within brackets has an unclear meaning: maybe a missing reference? Please check.
Line 426: please check the question mark at the end of the line.
Line 480: a closing bracket is missing.

Experimental design

I have some points regarding the very interesting discussion of the OGC SE and related standards.
You well clarify some weaknesses of the OGC standard, but the sentence at line 269 (“ambigous explanation in the specification”) could be better supported at least by one example, that I would not assume to be the possibility to obtain the same result by two different functionalities, on the contrary I would expect an example of two different outcomes driven by the same usage of the standard.
Test suite: a very important aspect is highlighted saying that a test suite is missing – lines 271, 272 -, but I would ask to better clarify if and how the current study could contribute to this point, that the same standard admits to be critical considering that the “Conformance verification of the output of a system implementing SE requires in most cases visual interpretation” (OGC SE, 05-077r4, Annex A).
Even if the request from different communities of alternative encodings to OGC standards is a fact (not confined to the cartographic community, see for instance, about SWE, the discussion about JSON encoding at http://www.ogcnetwork.net/node/1646), I would ask you some reference to support your comments in this direction at line 316, that could serve to better identify the actual gaps and be of interest to the reader.

Validity of the findings

In section “New symbolization capabilities”, the reader could benefit by some indication of the gap between the actual implementation of SE in contrast to the new symbolization capabilities you are discussing. For example Unit Of Measure suggested by OGC are confined to “metre”, “foot”, “pixel”; Transformations are actually (if I’m not wrong) just Displacement (=Translation?), Rotation and Scale, while other affine transformations are missing; etc.

---

## Round 0.2 · accepted · Accept

The revised version complies to the recommendations of the reviewers so the paper can now be accepted.